# An Apple Fungal Infection Detection Model Based on BPNN Optimized by Sparrow Search Algorithm

**DOI:** 10.3390/bios12090692

**Published:** 2022-08-28

**Authors:** Changtong Zhao, Jie Ma, Wenshen Jia, Huihua Wang, Hui Tian, Jihua Wang, Wei Zhou

**Affiliations:** 1Mechanical Electrical Engineering School, Beijing Information Science and Technology University, Beijing 100192, China; 2Institute of Quality Standard and Testing Technology, Beijing Academy of Agriculture and Forestry Sciences, Beijing 100097, China; 3Department of Food and Bioengineering, Beijing Vocational College of Agriculture, Beijing 102206, China; 4Hebei Food Safety Key Laboratory, Hebei Food Inspection and Research Institute, Shijiazhuang 050091, China

**Keywords:** electronic nose, fungal infection, sparrow search algorithm, apples

## Abstract

To rapidly detect whether apples are infected by fungi, a portable electronic nose was used in this study to collect the gas information from apples, and the collected information was processed by smoothing filtering, data dimensionality reduction, and outlier removal. Following this, we utilized K-nearest neighbors (KNN), random forest (RF), support vector machine (SVM), a convolutional neural network (CNN), a back-propagation neural network (BPNN), a particle swarm optimization–back-propagation neural network (PSO-BPNN), a gray wolf optimization–backward propagation neural network (GWO-BPNN), and a sparrow search algorithm–backward propagation neural network (SSA-BPNN) model to discriminate apple samples, and adopted the 10-fold cross-validation method to evaluate the performance of each model. The results show that SSA can effectively optimize the performance of the BPNN, such that the recognition accuracy of the optimized SSA-BPNN model reaches 98.40%. This study provides an important reference value for the application of an electronic nose in the non-destructive and rapid detection of fungal infection in apples.

## 1. Introduction

Apples are one of the most popular fruits, rich in vitamins and various minerals, and widely grown all over the world. Preliminary studies have shown that eating apples regularly can reduce the risk of colon cancer, prostate cancer, and lung cancer, while apple peel also contains a plethora of indeterminate phytochemicals that have antioxidant properties [1]. In 2021, the global production of apples was about 76.1 million tons, and normal apples can usually be stored for about a year after picking. However, apples are susceptible to some fungal infections during storage and transportation, including *Aspergillus niger* [2], *Penicillium expansum* [3] and *Penicillium crustosum* [4]. These fungi can cause widespread corruption of apples if not detected and handled in time, which will bring about huge direct economic losses for farmers. To reduce the economic loss caused by fungal infection, it is necessary to design a fast, convenient, and safe method to detect whether apples are infected by fungi, so as to improve the overall status of the apple industry.

When the volatile gas of apples encounters gas sensors, specific reactions occur, and the characteristic response spectra of gas information are given. Gas chromatography–mass spectrometry (GC-MS) is a widely used analytical method, which has the advantages of high selectivity, small sample requirement, and high resolution. GC-MS has developed rapidly in the applications of food safety, industrial detection, environmental protection, and other fields, while showing advantages, such as high resolution and high sensitivity. Berrada et al. measured patulin in apple juice using GC-MS and investigated the effect of patulin on the stability of apples during storage [5]. Thin-layer chromatography (TLC) is a chemical analysis method that is widely used in qualitative analysis in the fields of food, medicine, and environment. TLC has a high detection efficiency and can complete an analysis in 10–20 min. High-performance liquid chromatography (HPLC) uses liquid as the mobile phase and uses a high-pressure system to effectively separate analytes. HPLC has the advantages of high sensitivity and a fast analysis speed and plays an important role in the field of food safety. It is also a globally recognized authoritative method for the qualitative detection of fungi [6]. However, although the above traditional methods can determine whether apples are infected by fungi, these processes need to be carried out in the laboratory, the detection process is complicated, operation by professionals is required, the cost of experiment is expensive, and real-time detection is not possible. Therefore, there is an urgent need for a fast and simple technique to determine whether the experimental sample is infected by fungi via the analysis of the volatile gas from the sample. The portable electronic nose is a fast, non-destructive, efficient, and chemical-free technology for analyzing volatile gas on-site [7]. In recent years, it has been used prominently, and is receiving increasing attention in the field of food safety and evaluation.

The convenient and fast operation of the electronic nose means it plays an important role in the quality inspection of fruits. Guo et al. collected the characteristic information of apples infected by different fungi through an electronic nose and used the BPNN pattern recognition model to classify and identify the information of apples; the results achieved were good [8]. Nouri et al. used an electronic nose combined with the BPNN pattern recognition model to detect the fungal infection of pomegranates, and the recognition rate of pomegranate samples infected with mycoplasma was as high as 100% [9]. This study shows that the electronic nose is a reliable and high-precision instrument for detecting the quality of pomegranate. Voss et al. used an electronic nose to capture the volatile gas of peaches to predict growth and maturity [10]. The results prove that the method of using an electronic nose combined with random forest (RF) can effectively predict the maturity date of peaches in orchards, thereby reducing farmers’ economic losses caused by neglect and late harvest. Yang et al. analyzed the volatile gas in yellow peaches through an electronic nose, accomplished non-destructive prediction of the compression damage degree of the fruit, discriminated the damaged fruit, and predicted the compression time, while the accuracy of identifying the damaged fruit was as high as 93.33% [11]. Guo et al. used an electronic nose combined with PCA-DA to predict the corruption area and corruption degree of apples and achieved a good result. The prediction accuracy of the corruption degree was 97.2%; this study proves that the electronic nose has a certain application value in the classification of corrupt apples and the quantitative detection of corrupt areas [12]. To date, several studies have reported the application of the electronic nose technology in fruit quality detection. Furthermore, regarding the optimization of the recognition model, Wu et al. used a sparrow search algorithm (SSA) to propose an evaluation model for predicting the economic losses suffered by subway stations after rainstorms and floods, which effectively solves the problems of the low efficiency and low prediction accuracy of traditional evaluation models [13]. This study proves that the support vector machine (SVM) and BPNN evaluation models optimized by SSA have higher accuracy and stability than other optimization algorithms and can effectively predict the economic losses of subway stations caused by rainstorms and floods. Jiang et al. proposed a method for detecting aflatoxin B1 content in wheat based on colorimetric sensor array technology and used the firefly algorithm to optimize sensor features with SSA to optimize the BPNN recognition model [14]. The result proves that the prediction accuracy and stability of the optimized BPNN recognition model have been improved, and the complexity of the model has been reduced. Some studies have shown that the electronic nose can detect fruit quality according to volatile gas components under different conditions, which has great potential applicability for fruit damage and spoilage detection. Electronic nose technology can strengthen the early rot inspection of fruits and reduce the economic losses of growers. However, few studies have used electronic noses combined with classification models to evaluate the effects of different fungi on apple spoilage. In addition, an electronic nose combined with the BPNN recognition model can often achieve better results; however, in the BPNN recognition model, the hidden layers are mainly responsible for modeling the complex functions of the network, and the number of nodes in the hidden layer has a great impact on the performance of the model, which may directly lead to the model falling into overfitting or underfitting. In the previous model building process, rich experience and continuous debugging were always required to find the appropriate number of nodes in the hidden layer. SSA is a new type of intelligent optimization algorithm that has been proposed in recent years; using SSA to optimize the number of nodes in the hidden layers of BPNN can help to quickly determine the appropriate number of nodes in each hidden layer, which precludes spending a lot of time and effort to debug the model manually while the performance of the optimized model is improved. We sought to detect conveniently, rapidly, and effectively fungal infection in apples, simplify the training process of the recognition model, and improve the performance of the recognition model. Therefore, this paper includes the following: (a) the use of the electronic nose to collect the volatile information of fresh apples, apples inoculated with *Aspergillus niger*, apples inoculated with *Penicillium expansum*, and apples inoculated with *Penicillium crustosum*; (b) the preprocessing of the collected data by filtering and removing outliers; (c) the dimensionality reduction of the preprocessed data; (d) finally, using the SSA-optimized BPNN model to classify apples infected with different fungi, comparing it with traditional pattern recognition methods.

## 2. Materials and Methods

### 2.1. Materials

The “Fuji” apples selected in this experiment came from apple plantations in Gansu Province, China. In total, 160 ripe apples were selected and randomly divided into 4 groups, 40 apples in each group, namely, Group A, Group B, Group C, and Group D. The fungi inoculated into the middle apples were *Aspergillus niger*, *Penicillium expansum*, and *Penicillium crustosum*. The apple samples were pretreated with 75% alcohol on a sterile bench and dried at room temperature. Then, four holes were punched in four directions in each apple in the three groups containing the inoculator (A, B, and C). Sample apples were inoculated with 7-day-old molds through drilled loops, and the holes were covered with sterile film. The mold-inoculated apples were then placed in a 1000 mL beaker, sealed with plastic wrap, and then placed in a 25 °C constant-temperature incubator for 5 days. Before the test, the apple samples were taken out of the incubator and left to rest for 30 min. To eliminate the influence of residual gas on the experimental results, the electronic nose was cleaned with inert gas before using. Electronic nose parameters were as follows: cleaning time 500 s, collection time 350 s, sampling interval 1 s, injection flow 150 mL/min.

The sensor array of the portable electronic nose in this experiment was composed of electrochemical sensors. The portable electronic nose was primarily composed of three parts: the control unit, the sensor room, and the data acquisition and transmission unit. The portable electronic nose used in the experiment is shown in Figure 1a, and the schematic diagram of the portable electronic nose is shown in Figure 1b. The sampling valve controls the gas entry into the sealed bottle, the injection valve controls the entry of gas into the air chamber and the flow of the gas, and the injection valve and the vacuum valve work together to prevent outside gas from entering the air chamber. The response curve of the No. 1 sensor C_2_H_4_-20 during the sampling process is shown in Figure 2. In addition, Fameview (V7.6.12.4) configuration software (Beijing Jiekong, Beijing, China) was used to collect and save the electronic nose data, while the Modbus protocol was used for data transmission and communication with the hardware, so as to permit human–computer interaction. In this experiment, two identical sensors (numbered 1 and 5) were selected as indicators to identify whether the collected data were abnormal. If the difference between the two 150–300 s sensors was greater than 1.2 mg/L, the specimen was considered anomalous and removed. Table 1 lists the sensor names and performance specifications, and Table 2 lists the specifications of the metal oxide sensors commonly used for PEN3 electronic noses. From the comparison between Table 1 and Table 2, it can be seen that the recognition accuracy of the electrochemical sensor is higher than that of the metal oxide sensor. In addition, the electronic nose system proposed in this paper has a lower cost and a higher detection accuracy than the electronic nose of the same price. It is also very convenient to carry; the volume of the electronic nose system is about 0.04 m^3^ and its weight is about 15 kg, which makes it convenient for inspection personnel as they carry out inspection and analysis on site. The operating platforms used in this experiment were PyCharm 2021.3 (JetBrains, Prague, Czech Republic), Tensorflow2 (Google, Menlo Park, CA, USA), and Matlab2018b (MathWorks, Portola Valley, CA, USA).

### 2.2. Methods

#### 2.2.1. Data Preprocessing

The portable electronic nose selected in this experiment is equipped with 8 sensors, and each sensor can detect different gas components. Since the sensor array of the portable electronic nose in this experiment used an electrochemical sensor, it has the characteristics of high sensitivity, but low response compared with metal oxide sensors, while the sensor has a certain cross-sensitivity (it can detect multiple gases). Based on this characteristic, we took the integral value, variance value, average differential value, maximum gradient value, relatively stable average value and energy value of the response curve of each sensor over 30–300 s as the characteristic information of the electronic nose, so the characteristic parameter of each sample is 48.

##### Smooth Filter

In this experiment, 3-point linear smoothing, 5-point linear smoothing, 7-point linear smoothing, 9-point linear smoothing, and 11-point linear smoothing algorithms were selected to remove noise from the data, and their results were compared. The response curves of each sensor after smoothing are shown in Figure 3. As can be seen from Figure 3, the response curve after 7-point smoothing is relatively smooth, and can maintain the basic characteristic information of the data, so the 7-point linear smoothing method was selected for the preprocessing operation in this experiment. In addition, the response curves of the 7NE/H_2_S-1000 and PID-300 sensors are always 0, because fresh apples and apples infected with fungi release less H_2_S gas. From Table 1, it can be seen that H_2_S-50 and H_2_S-1000 are both effective sensors for detecting H_2_S gas; the difference between them is that the detection precision of H_2_S-50 is much higher than that of H_2_S-1000. It can be seen from Figure 3a that the response curve of the H_2_S-50 sensor is less than 1, which means that the sample releases less H_2_S gas, and has not reached the minimum detection range of H_2_S-1000. Therefore, the response curve of H_2_S-1000 is always 0. Similarly, the concentration of VOC gas released by the sample in the sealed bottle did not reach the minimum detection range of the PID-300 sensor, so the response curve of the PID-300 sensor is always 0. As such, in this study, we only analyzed the data measured by 6 sensors, excluding 7NE/H_2_S-1000 and VOC-300, so the characteristic parameter of each sample is 36.

##### Eliminate Outliers

Due to the performances of the sensors and the effects of the external environment, the collected sample data may contain abnormal values, and abnormal data will directly affect the accuracy and stability of the recognition model. In order to eliminate this effect, Mahalanobis distance was used to remove abnormal data from the original data. Mahalanobis distance is a method used for calculating the distance between points and distribution and was proposed by Indian statistician P. C. Mahalanobis. The inconsistency and correlation between the scales of each dimension can be used to effectively evaluate the similarity within the data. Sun et al. used Mahalanobis distance combined with Monte Carlo cross-validation to effectively remove outliers from the hyperspectral data of tobacco leaf water content [15].

In this experiment, Mahalanobis distance and the method of difference judgment for the No. 1 and No. 5 sensors mentioned above were used to eliminate 30 abnormal sample data: 8 apples inoculated with *Penicillium crustosum*, 6 apples inoculated with *Aspergillus niger*, 5 apples inoculated with *Penicillium expansum*, and 11 fresh apples. Then, the KNN, SVM, and BPNN models were trained with the original data and the data after removing outliers, respectively, and the 10-fold cross-validation method was used to evaluate the performance of each model; the results are shown in Table 3. It can be seen that the accuracy and stability of each pattern recognition model were improved after removing outliers from Table 3. The results show that removing outliers from the raw data can effectively improve the performance of the recognition model.

##### Data Dimensionality Reduction

Principal component analysis (PCA), factor analysis (FA), and linear discriminant analysis (LDA) are common data dimensionality reduction methods. The purpose of data dimensionality reduction is to reduce the dimension of the original data, remove useless information from the original data, and increase the recognition accuracy of the recognition model on the premise of retaining as much as possible of the feature information of the original data. PCA, FA, and LDA were performed on the sample data after removing outliers, and the results are shown in Figure 4. The 10-fold cross-validation method was used to evaluate the performances of the SVM, BPNN, and KNN recognition models with different dimensionality reduction methods. The results are shown in Table 4. The results show that the recognition accuracies of the models after PCA, FA, and LDA were improved, and the dimensionality reduction performance of LDA was the best among the three methods.

#### 2.2.2. Pattern Recognition Model

In recent years, with the rapid development of machine learning, the combination of electronic noses and pattern recognition models such as KNN, RF, SVM, CNN, and BPNN in machine learning has increased the prominence of the electronic nose in many fields. Virtanen et al. successfully identified five common pathogenic bacteria of acute sinusitis by combining the electronic nose and KNN while providing a pathological basis for the treatment of acute sinusitis [16]. Tian et al. made full use of the advantages of high RF stability, short time consumption, and high precision, and proposed an electronic nose and RF model based on the rapid detection of yogurt flavor acceptability. The research proved that the combination of an electronic nose and RF can be used to effectively evaluate the acceptability of yogurt flavor [17,18]. Jiang et al. used a combination of the electronic nose and SVM to classify five common odors [19]. Kang et al. uses CNN to process electronic nose data based on metal oxide sensor arrays to achieve the real-time detection of CO, NH_3_, NO_2_, CH_4_, and C_3_H_6_O gases [20]. Gu et al. used a combination of the electronic nose and BPNN to detect early Aspergillus in rice, and the recognition accuracy reached 96.40% [21].

To improve the performance of the pattern recognition model, an optimization algorithm can be used. Currently, the most commonly used optimization algorithms are particle swarm optimization (PSO) [22], gray wolf optimization (GWO) [23], and sparrow search algorithm (SSA) [24]. These are inspired by the feeding behaviors of animals in nature. SSA is a recently proposed swarm intelligence optimization algorithm. The sparrow population in SSA is divided into two parts: producers and scroungers. Producers have a high fitness value and energy reserve, and their main task is to provide scroungers with directions and areas for foraging. The search range of producers is larger than that of scroungers. Scroungers follow producers to find food and obtain their own energy reserve, and thus increase their fitness value, and some scroungers continuously increase their own energy reserves through predation, thus turning themselves into producers. In addition, some sparrows in the sparrow population will act as forewarners. Forewarners will issue a warning signal when danger is coming, and at the same time spread into the safe area to obtain a better position. When the alarm value is greater than the set threshold, the producers will lead all scroungers out of the danger zone. A schematic diagram of SSA is shown in Figure 5. SSA has better global search and local development capabilities and can consider all the variable factors of the population, so that the population can quickly move into the optimal position. SSA also has the advantages of fewer iterations and higher prediction model accuracy.

## 3. Results

In order to verify the abilities of the above eight recognition models to recognize fungus-infected apples, a multi-algorithm pattern recognition platform was developed using the PyQt5 tool in this experiment. The user interface enables human–computer interaction through the mouse and keyboard to adjust the parameters of different recognition models. As shown in Figure 6a, users can select different pattern recognition models on the main interface of the platform and enter the corresponding recognition model interface by clicking the button. Then, as shown in Figure 6b–i, the user can adjust the parameters of the current recognition model according to the label prompts, select training sample data to train the model, and assess the performance of the model. When the training of the recognition model is completed, the inspectors can begin to inspect the apple samples. First, one must put the apple sample into a sealed bottle to collect its electronic nose data, select the recognition model that has been trained in the multi-algorithm pattern recognition platform, and finally select the collected electronic nose data for this model to use to detect the apple sample. The result will soon appear in the recognition result display area below, as shown in Figure 6j. The whole process does not require the use of any chemical reagents, nor will it cause damage to the apple samples, and the testing process will not affect the edibility and sales of the apple samples.

### 3.1. The Recognition Accuracy of the Model Optimized by SSA-BPNN Is Higher

The characteristic information of apples inoculated with three different fungi and fresh apples was collected by the electronic nose, and the collected characteristic information was preprocessed. Then, we used the above-mentioned multi-algorithm pattern recognition platform to train each recognition model while using the 10-fold cross-validation method to evaluate the performance of each model; the results are shown in Figure 6b–i, and the summarized results are shown in Table 5. It can be seen from Table 5 that CNN, RF, KNN, SVM, and BPNN were used for identifying the apples inoculated with *Aspergillus niger*, *Penicillium expansum*, and Penicillium officinale, as well as fresh apples. The average accuracy values of each recognition model are 57.80%, 86.92%, 89.23%, 91.07%, and 93.17%, and the standard deviation of the accuracy is less than 0.08. In terms of training time, although the training times of the RF, KNN, and SVM recognition models are shorter, they are not as good as the BPNN model in terms of recognition accuracy. Following this, three optimization algorithms, PSO, GWO, and SSA, were used to optimize the BPNN model. The number of iterations was set to 100 and the number of populations was 15; the accuracy of the BPNN model was used as the fitness function, and the 10-fold cross-validation method was also used to evaluate the performance of the model after optimization. Then, the average recognition accuracy values of the PSO-BPNN, GWO-BPNN, and SSA-BPNN models are 94.62%, 96.16% and 98.40%, respectively, and the standard deviations of the accuracy are 0.091, 0.064 and 0.032, respectively. The training times are 8936.21 s, 8723.43 s and 9193.02 s, respectively. In addition, as regards the evaluation score of the true positive rate (TPR) and the F1 score of each recognition model, the scores of SSA-BPNN are higher than those of other recognition models. In summary, the experimental results show that the SSA-BPNN recognition model proposed in this paper achieves outstanding performance in detecting fungal infection in apples.

### 3.2. SSA-BPNN Has Faster Convergence

The variations in the fitness functions of SSA, GWO, and PSO with the number of iterations in the optimization process are shown in Figure 7. Although SSA-BPNN has no obvious beneficial effect on the optimization time required for 100 iterations, it can be seen from Figure 7 that PSO-BPNN, GWO-BPNN, and SSA-BPNN enable the model to reach the optimal state after 81, 70, and 36 iterations, respectively, which shows that the convergence speed of SSA is higher than those of PSO and GWO, and it has a better optimization capacity. Compared with the GWO-BPNN and PSO-BPNN models, the SSA-BPNN model proposed in this study has obvious advantages in terms of recognition accuracy, stability, and convergence speed. 

SSA is an intelligent optimization algorithm that has been proposed in recent years. Using SSA to optimize the BPNN recognition model can help the BPNN model to quickly find the optimal parameters thus avoiding the need to spend a lot of time and energy to debug the recognition model manually. With regard to the huge economic losses caused by fungal infection in the process of storage and transportation, the portable electronic nose combined with the SSA-BPNN method proposed in this study can effectively detect and identify common fungi and take preventive measures in time. The measures can effectively reduce the economic losses caused by fungal infection in apples.

## 4. Discussion

Apples are loved because of their delicious taste and rich nutrition. The annual global demand and supply of apples are huge, but some apples will inevitably be infected by fungi during the storage process, which will introduce huge economic losses to apple merchants. Electronic noses can be used to detect volatile substances in contact with the sensor array, which facilitates the quality detection of fruits. With the continuous development and progress of machine learning, the combination of electronic noses and machine learning provides a fast, non-destructive and easy-to-operate method for fruit quality detection and has gradually attracted people’s attention. Compared with machine learning algorithms such as KNN, RF, CNN, and SVM, BPNN has the best performance. However, BPNN has many parameters of note; in particular, a small change in the number of nodes in the hidden layers can easily affect the overall performance of the BPNN network model. The selection of optimal parameters for BPNN has always been the most important task in the process of model building. On this basis, this paper proposed to use SSA to optimize the BPNN network model by finding the optimal parameters of the hidden layers of BPNN through continuous iteration, such that the model can quickly reach the optimal state. The TPR, F1 score, and accuracy of the model’s recognition results after optimization achieved 97.31%, 0.976%, and 98.40%, respectively—4.56%, 0.047%, and 5.23% higher than those before optimization. The reason for this is the lower number of iterations compared to PSO and GWO. However, when the electronic nose is used to detect the quality of apples, its performance will be affected by external factors. Interference outside the normal range will give rise to abnormalities in the characteristic information of the samples collected by the electronic nose, which will eventually lead to unreliable test results. In order to improve the reliability of the detection results, in our follow-up research, we will combine the electronic nose with other detection equipment and collect characteristic information from the samples at the same time, using the method of data fusion to detect and identify the samples.

## Figures and Tables

**Figure 1 biosensors-12-00692-f001:**
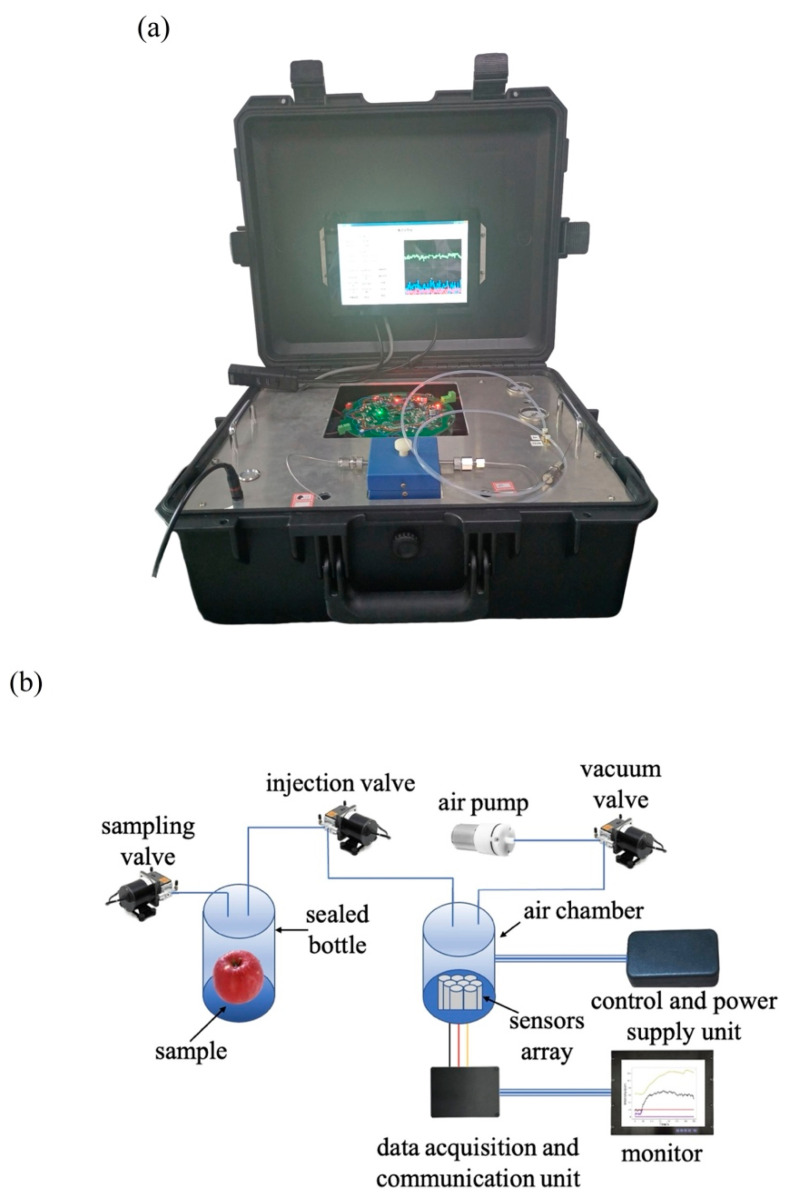
(**a**) The portable electronic nose; (**b**) schematic diagram of the portable electronic nose.

**Figure 2 biosensors-12-00692-f002:**
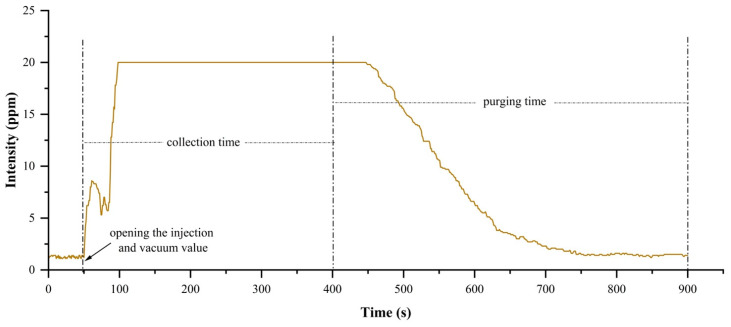
The response curve of the No. 1 sensor C_2_H_4_-20 during the entire sample-collection period. When opening the injection and vacuum valve at 50 s, the collection time lasts 350 s, and the purging of the electronic nose lasts 500 s.

**Figure 3 biosensors-12-00692-f003:**
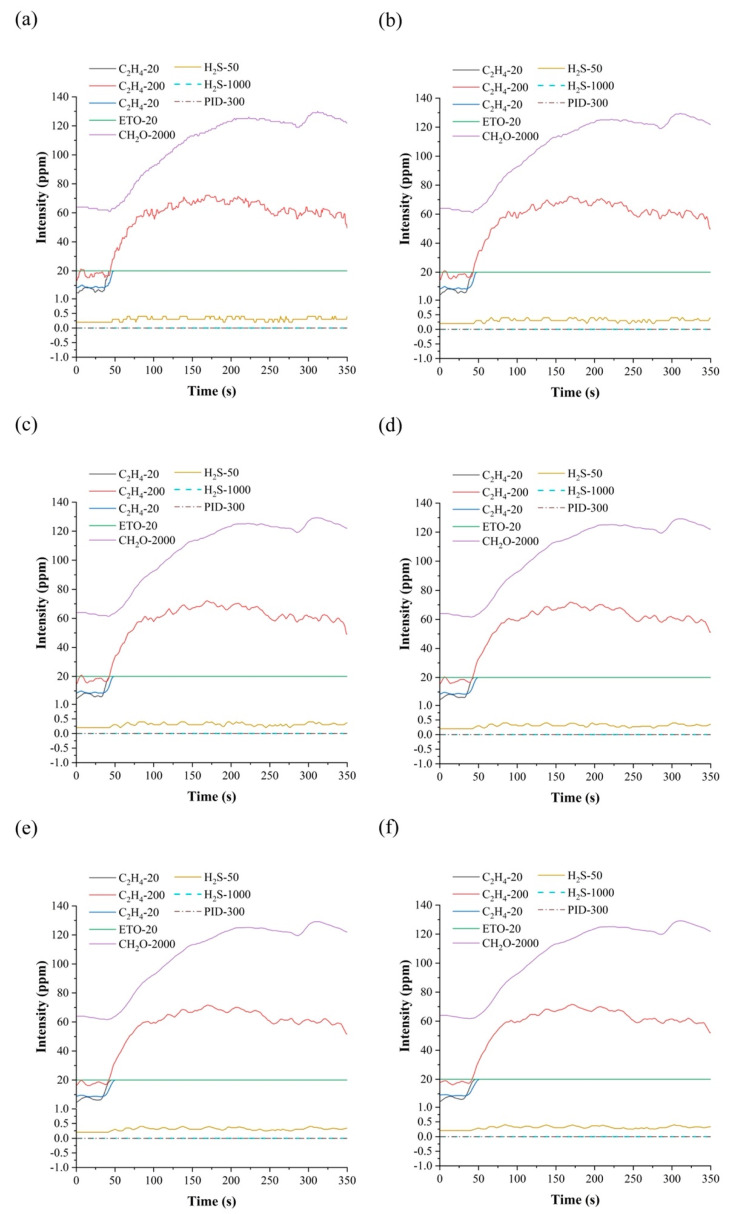
(**a**) Original response curve. (**b**) Response curve after 3-point smoothing filtering. (**c**) Response curve after 5-point smoothing filtering. (**d**) Response curve after 7-point smoothing filtering. (**e**) Response curve after 9-point smoothing filtering. (**f**) Response curve after 11-point smoothing filtering.

**Figure 4 biosensors-12-00692-f004:**
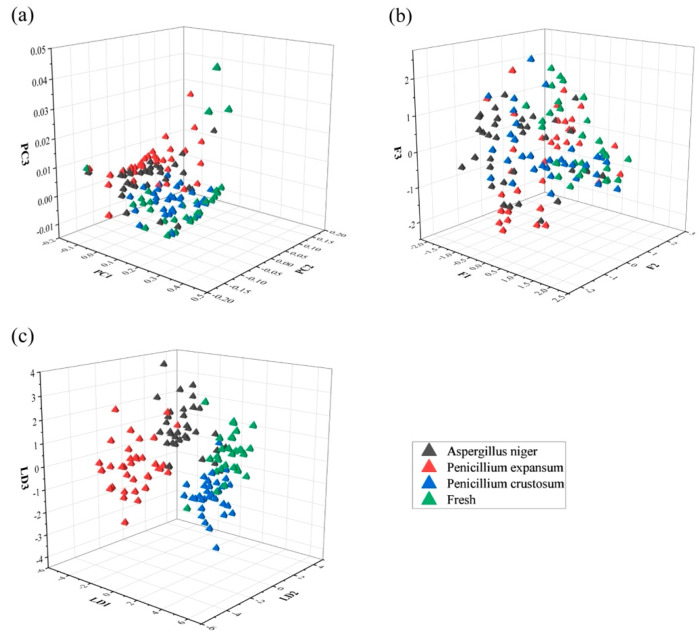
Comparison of different data dimensionality reduction methods: (**a**) PCA dimensionality reduction analysis; (**b**) FA dimensionality reduction analysis; (**c**) LDA dimensionality reduction analysis.

**Figure 5 biosensors-12-00692-f005:**
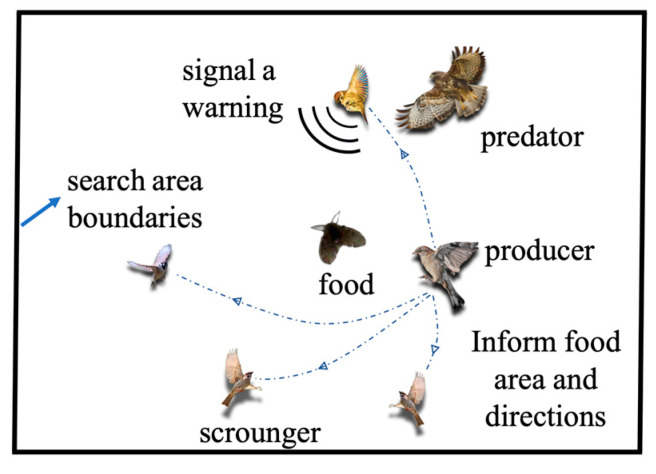
Schematic diagram of the sparrow search algorithm.

**Figure 6 biosensors-12-00692-f006:**
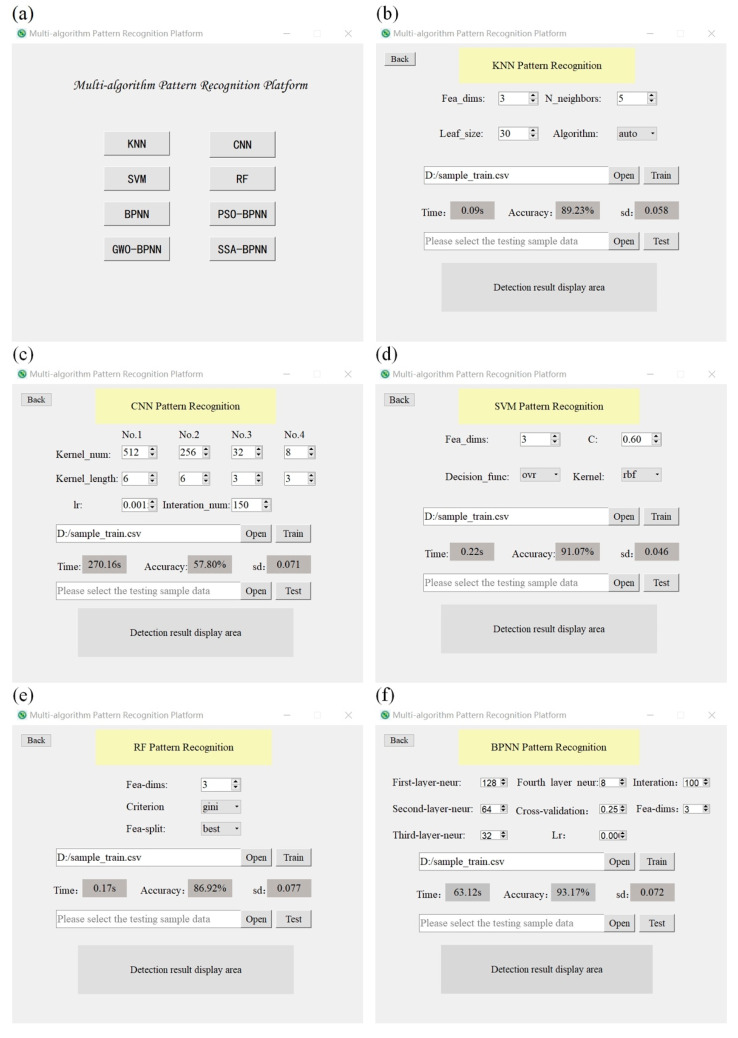
Multi-algorithm pattern recognition model platform. (**a**) Main interface of multi-algorithm pattern recognition platform. (**b**) KNN pattern recognition interface. (**c**) CNN pattern recognition interface. (**d**) SVM pattern recognition interface. (**e**) RF pattern recognition model interface. (**f**) BPNN pattern recognition model interface. (**g**) PSO-BPNN pattern recognition interface. (**h**) GWO-BPNN pattern recognition interface. (**i**) SSA-BPNN pattern recognition interface. (**j**) SSA-BPNN detection sample interface.

**Figure 7 biosensors-12-00692-f007:**
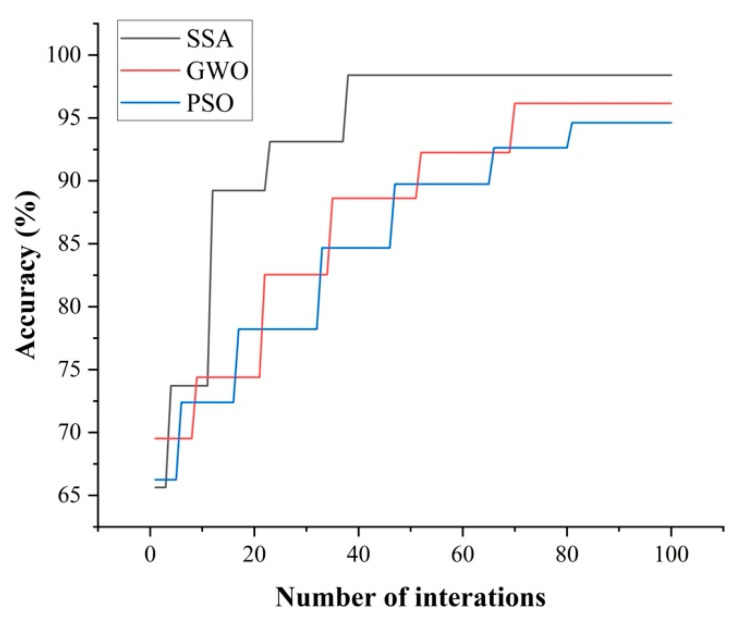
Variation trend of the fitness function curves of SSA, GWO, and PSO with the number of iterations. The SSA-BPNN model reached the optimal state first.

**Table 1 biosensors-12-00692-t001:** Types of gas-sensitive sensors in the portable electronic nose sensor array, and their sensitive gas and detection accuracies.

Sensor Number	Sensor Model	Sensitive Gas	Detection Precision (ppm)
1	7NE/C_2_H_4_-20	C_2_H_4_	0.4
2	7NE/H_2_S-50	H_2_S	1
3	7NE/H_2_S-1000	H_2_S	20
4	7NE/C_2_H_4_-200	C_2_H_4_	4
5	7NE/C_2_H_4_-20	C_2_H_4_	0.4
6	7NE/ETO-20	C_2_H_4_O	0.4
7	7NE/PID-300	VOC	6
8	7NE/CH_2_O-2000	CH_2_O	40

**Table 2 biosensors-12-00692-t002:** Names of sensors in the PEN3 electronic nose sensor array, their sensitive gas and detection accuracies.

Sensor Name	Sensitive Features	Representative Gas and Detection Precision (ppm)
W1C	Sensitive to aromatic compounds	Methylbenzene, 10
W3C	Aromatic compounds, particularly sensitive to ammonia	Benzene, 10
W5C	Aromatic compounds such as alkanes and compounds with relatively small polarity	Propane, 1
W1S	Particularly sensitive to methane contained in specimens	Methane, 100
W2S	Particularly sensitive to ethanol contained in specimens	Carbon monoxide, 100
W3S	Sensitive to high-concentration alkanes, especially methane, in specimens	Methane, 100
W5S	Sensitive to nitrogen oxides, extremely sensitive to negatively charged nitrogen oxides	Nitrogen dioxide, 1
W6S	Only detects hydrogen	Hydrogen, 100
W1W	Mainly sensitive to sulfides, also sensitive to organic sulfides	Hydrogen sulfide, 1
W2W	Mainly sensitive to aromatic compounds and organic sulfides	Hydrogen sulfide, 1

**Table 3 biosensors-12-00692-t003:** Average accuracy and standard deviation of KNN, BPNN, and SVM pattern recognition models before and after removing outliers.

Model	Sample Set	Validation Set Accuracy	Standard Deviation of Accuracy
KNN	before removing	62.37%	0.326
	after removing	68.29%	0.163
BPNN	before removing	56.38%	0.452
	after removing	69.21%	0.263
SVM	before removing	57.32%	0.318
	after removing	64.65%	0.227

**Table 4 biosensors-12-00692-t004:** Average accuracy of KNN, BPNN, and SVM pattern recognition models after different dimensionality reduction methods; the LDA dimensionality reduction method is the best among the three dimensionality reduction methods.

Dimensionality Reduction Method	Model	Average Accuracy	Standard Deviation of Accuracy
	SVM	64.65%	0.227
None	BPNN	69.21%	0.263
	KNN	68.29%	0.163
	SVM	66.29%	0.138
PCA	BPNN	75.38%	0.149
	KNN	73.85%	0.104
	SVM	65.50%	0.136
FA	BPNN	72.14%	0.147
	KNN	63.53%	0.119
	SVM	91.07%	0.046
LDA	BPNN	93.17%	0.072
	KNN	89.23%	0.058

**Table 5 biosensors-12-00692-t005:** TPR, F1 score, average accuracy, standard deviation of accuracy, and training time of different pattern recognition models by 10-fold cross-validation.

Preprocessing	Model	TPR	F1 Score	Average Accuracy	Standard Deviation of Accuracy	Training Time (s)
\	CNN	61.03%	0.564	57.80%	0.071	270.16
LDA	RF	84.36%	0.836	86.92%	0.077	0.17
LDA	KNN	88.12%	0.878	89.23%	0.058	0.09
LDA	SVM	90.69%	0.903	91.07%	0.046	0.22
LDA	BPNN	92.75%	0.929	93.17%	0.072	63.12
LDA	PSO-BPNN	93.82%	0.937	94.62%	0.091	8936.21
LDA	GWO-BPNN	95.83%	0.951	96.16%	0.064	8723.43
LDA	SSA-BPNN	97.31%	0.976	98.40%	0.032	9193.02

## Data Availability

The raw and processed data presented in this study are publicly available on Figshare at https://doi.org/10.6084/m9.figshare.19759120.v1 (accessed on 13 May 2022) and can be cited. The code in the study is publicly available on GitHub at https://github.com/palenn/Identifying_fungal-infected_apples.git (accessed on 13 May 2022) and can be cited.

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
