# Peer review of "An Apple Fungal Infection Detection Model Based on BPNN Optimized by Sparrow Search Algorithm"

_biosensors, 2022, doi:10.3390/bios12090692_

Round 1

Reviewer 1 Report

Need serious English editing before publishing.

Line 84, use first author et. al. when citing other people’s work unless the paper only has one author.

Please include a photo of the electronic nose in Figure 1.

Table 1 and 2 should  use different units

Figure 2 there are 8 curves in the legend but only 4-5 curves are visible in the plot. Why is C2H4O-20 curve flat?

Reviewer 2 Report

The topic is interesting, but it requires some major improvements before considering it valid for a journal publication. The major and minor concerns are as follows:

1.     Graphical abstract can be made more attractive.

2.     The picture of the developed E-Nose must be included in the manuscript.

3.     Page 3, line 108 to 112 is unclear.

4.     The process is reported as non-destructive. The authors should clarify the process in more detail.

5.     Some comments on redundant sensors must be given.

6.     The response curves in Fig. 2 are incomplete. A new set of response curves clearly indicating the initial action (settling time), sensing and purging must be included.

7.     In Table 3, the test set accuracy is confusing.

8.     Sub-section 2.2.2 gives only basic definitions. The sub-section can be omitted.

9.     Table 5 should include more performance parameters of the models used.

Round 2

Reviewer 1 Report

Still need serious editing of English language

Author Response

Thank you very much for your valuable suggestions for our manuscript in your busy schedule, based on your revision suggestions, We have carefully revised the grammar and language in the manuscript. 

Author Response

This manuscript is a resubmission of an earlier submission. The following is a list of the peer review reports and author responses from that submission.

Round 1

Reviewer 1 Report

he authors developed apple fungal infection detection model based on BPNN optimized by sparrow search algorithm. The work is interesting in Engineering point of view, and it is helpful for the biosensors in food processing. I suggest accepting the work after minor revisions for some points, as follows:

1.       The author should show the baseline curve for sensor array in only air to confirm that the data shown in Fig.2 is coming mainly from Apple spoilage.

2.       Mention the type of data acquisition used here.

3.       In the sentence in line 142, “   .. the specimen will then be considered anomalous and removed” is not clear.

4.       to be fair in comparison of sensors, Table 2 should have the same parameters of Table 1. What is the relation between the sensor mentioned Table 1 and apple spoilage detection?

5.       Enhance Figure 2 quality, I see Fig 2a is enough to show.

6.       What does “abnormal data” mean here?

7.       Conclusion part is missing. Please revise 4. Section